# Artificial intelligence model for predicting sexual dimorphism through the hyoid bone in adult patients

**Aline Xavier Ferraz**[1], **Ângela Graciela Deliga Schroder**[2], **Flavio Magno Gonçalves**[1], **Erika Calvano Küchler**[3], **Rosane Sampaio Santos**[1], **Bianca Simone Zeigelboim**[1], **Ana Paula Testa Pezzin**[4], **Karinna Verissimo Taveira**[5], **Allan Abuabara**[4], **Flares Baratto-Filho**[4]*, **Cristiano Miranda de Araujo**[1]

1 Postgraduate Program in Human Communication Health, Tuiuti University of Paraná, Curitiba, Paraná, Brazil, 2 School of Dentistry, Tuiuti University of Paraná, Curitiba, Brazil, 3 Department of Orthodontics, Medical Faculty, University Hospital Bonn, Bonn, Germany, 4 University of the Region of Joinville (Univille), Joinville, Santa Catarina, Brazil, 5 Department of Morphology- Center of Biosciences, Federal University of Rio Grande do Norte, Natal, Rio Grande do Norte, Brazil

* fbaratto1@gmail.com

**Data Availability Statement:** All relevant data are within the manuscript.

**Funding:** The author(s) received no specific funding for this work.

## Abstract

The objective of this study was to develop a predictive model using supervised machine learning to determine sex based on the dimensions of the hyoid bone. Lateral cephalometric radiographs of 495 patients were analyzed, collecting the horizontal and vertical dimensions of the hyoid bone, as well as the distance from the hyoid to the mandible. The following algorithms were trained: Logistic Regression, Gradient Boosting Classifier, K-Nearest Neighbors (KNN), Support Vector Machine (SVM), Multilayer Perceptron Classifier (MLP), Decision Tree, AdaBoost Classifier, and Random Forest Classifier. A 5-fold cross-validation approach was used to validate each model. Model evaluation metrics included areas under the curve (AUC), accuracy, recall, precision, F1 score, and ROC curves. The horizontal dimension of the hyoid bone demonstrated the highest predictive power across all evaluated models. The AUC values of the different trained models ranged from 0.81 to 0.86 on test data and from 0.78 to 0.84 in cross-validation, with the random forest classifier achieving the highest accuracy rates. The supervised machine learning model showed good predictive accuracy, indicating the model's potential for sex determination in forensic and anthropological contexts. These findings suggest that the application of artificial intelligence methods can enhance the accuracy of sex estimation, contributing to significant advancements in the field.

## Introduction

Human sexual dimorphism is a vast field of study that encompasses various biological and psychological characteristics [1]. Sex identification is a primary goal in forensic science, essential in situations such as mass disasters, explosions, or wars [2]. A biological marker of human

**Competing interests:** The authors have declared that no competing interests exist.

identity is bone structure, extensively investigated as a dimorphic trait [1]. Research across different populations has sought to identify sex differences, especially through analysis of craniofacial structures [3, 4]. In this context, bones of the head and neck are important due to their high resilience. Measurements of craniofacial bones, such as the frontal bone, mandible, frontal and maxillary sinuses, cervical vertebrae, and hyoid bone, have been used for sex determination in forensic science [2, 5–9]. The hyoid bone, in particular, is highly dimorphic and holds significant utility in forensic contexts [10].

The hyoid bone is a U-shaped craniofacial bone located on the ventral aspect of the neck at the level of the fourth cervical vertebra. It connects the muscles, ligaments, and fascia of the pharynx, jaw, and skull [11–13]. Although the hyoid bone is not articulated with any other bone, to maintain the airway, swallowing, preventing regurgitation, and supporting head posture without the hyoid would be impossible [11]. This bone can exhibit morphological sexual differences, in which men have a slightly larger and more robust hyoid bone, while women have smaller dimensions [14]. Although these morphological distinctions are slight, they are relevant in forensic and anthropological contexts to aid sex determination of human remains, complementing other bone analyses and scientific methods [13, 15, 16].

Data science techniques, such as machine learning, have been employed to estimate sex in forensic science [5, 17, 18]. Machine learning, a branch of artificial intelligence, makes predictions without being explicitly programmed to do so, using mathematical models derived from a training dataset [19]. Previous studies have employed artificial intelligence-based methods to determine sexual dimorphism using the unfused hyoid bone and 3D computed tomography images of Indian and Turkish populations, showing good predictive capacity for these groups [20, 21]. However, the literature on the application of machine learning to the hyoid bone remains limited, with few studies focusing on specific populations and methods. Notably, there is a lack of studies using supervised machine learning methods to determine sex in adults based on hyoid bone measurements from two-dimensional images, particularly within Brazilian populations. Thus, the aim of this study was to develop a predictive model using supervised machine learning to predict sex based on hyoid bone dimensions.

## Materials and methods

### Study design and setting

This cross-sectional study screened patients who underwent lateral cephalometric radiography exams at a private clinic located in the southern region of Brazil. The research was conducted in accordance with the principles established in the Helsinki Declaration and received approval from the Research Ethics Committee of Tuiuti University of Paraná, Brazil (Approval Number: 6,305,456). The study obtained permission to access data with waived informed consent, due to the retrospective nature of the analysis, provided that patient identification was not disclosed. Therefore, all data were anonymized prior to access. Data collection took place in May 2024.

### Participants

Adults of both sexes, aged 18 to 59 years, with no history of surgeries in the mandibular or hyoid area were included. Children, adolescents, elderly individuals, edentulous individuals or those lacking posterior teeth, as well as those using full dentures, removable/partial posterior dentures, or any device that could affect mandibular position, were excluded. Patients with syndromes, history of orthognathic surgery, or with condylar and/or mental prostheses were also excluded.

## Variables and measurements

Data collection was conducted by a single examiner previously calibrated by a senior expert with a master's degree in dental radiology. Additionally, intra-examiner calibration was performed and the same evaluator repeated the measurements after a 14-days interval. The intra-class correlation coefficient (ICC) was calculated using the statistical software Jamovi (Jamovi Project version 2.3.6.0), resulting in values > 0.8 for both analyses.

Measurements of the hyoid bone were performed using linear tracings on images derived from lateral cephalometric radiographs (15, 16). These radiographs were obtained using the Orthophos XGS 2D/3D equipment from Dentsply Sirona (Dentsply Sirona Inc, New York, USA), with an exposure time of 14.9 seconds, 77 kV, and 14 mA. The acquisition followed the recommended technical guidelines: the Median Sagittal Plane (MSP) was perpendicular to the ground, the Frankfurt Plane was parallel to the ground, and the patient was positioned in the cephalostat with maximum occlusion and lips at rest. The following variables were collected:

1. Horizontal length of the hyoid bone (HL)—Total length of the hyoid bone, measured from the most posterior point of the greater horn to the most anterior point of the body of the hyoid bone;

2. Vertical length of the hyoid bone (VL)—Total height of the hyoid bone, measured at the central point between the posterior point of the greater horn and the most anterior point of the body;

3. Hyoid to mandible distance (DHM)—Distance between the most posterior and superior point of the greater horn of the hyoid bone and the Gonion point, located at the bisector of the angle formed by the posterior plane of the ramus and the mandibular plane;

The points used for measuring each variable can be visualized in the Fig 1.

To determine the required sample size, a pilot study was conducted to calculate the effect size of each variable, considering the comparison of hyoid bone dimensions between male and female. In the sample size calculation, the variable with the smallest effect size, which requires a larger number of participants, was considered. A statistical power $(1 - \beta)$ of 80%, a two-tailed test, and an effect size of 0.253 were established. Based on these parameters, it was determined that the minimum required number of participants would be 494 patients.

## Data analysis and model building

A univariate analysis was conducted using the independent samples t-test to compare the mean values of the collected variables with respect to sex (dependent variable). The test power for each comparison was calculated using the statistical software GPower, version 3.1.9.6. Effect size was determined by calculating Cohen's d. A significance level of 5% ($\alpha = 0.05$) was adopted to allow the inclusion of any relevant variable in the model.

The following supervised machine learning algorithms were used in constructing the predictive model: Logistic Regression, Gradient Boosting Classifier, K-Nearest Neighbors (KNN), Support Vector Machine (SVM), Multilayer Perceptron (MLP) Classifier, Decision Tree, Adaboost Classifier, and Random Forest Classifier. Model optimization was performed using the Grid Search method to find the best combination of hyperparameters for each model. This technique involves systematically evaluating predefined combinations of different hyperparameters. The Python programming language was used to build all models, utilizing the 'scikit-learn' library. The code used is openly accessible (DOI: 10.5281/zenodo.12518509).

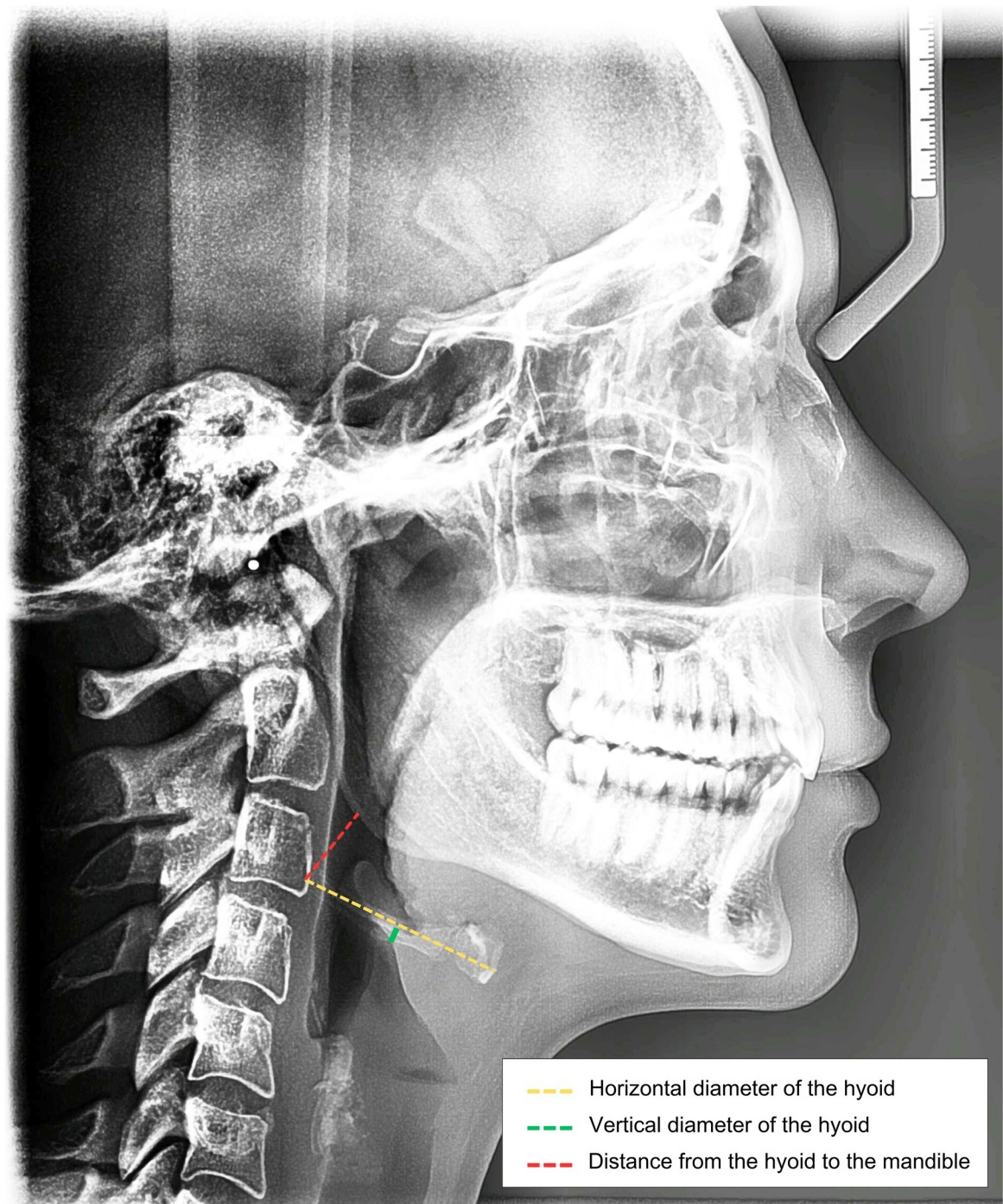

Legend:
- - - - Horizontal diameter of the hyoid
- - - - Vertical diameter of the hyoid
- - - - Distance from the hyoid to the mandible

**Fig 1. Variables collected for hyoid bone measurement in lateral cephalometric radiograph examination.**

### Training, cross-validation, and test

The data were divided into training and cross-validation sets (70%) and a test set (30%), used exclusively to evaluate the predictive capability of the model. To ensure class balance in the predictions, the Synthetic Minority Over-sampling Technique (SMOTE) was applied to the

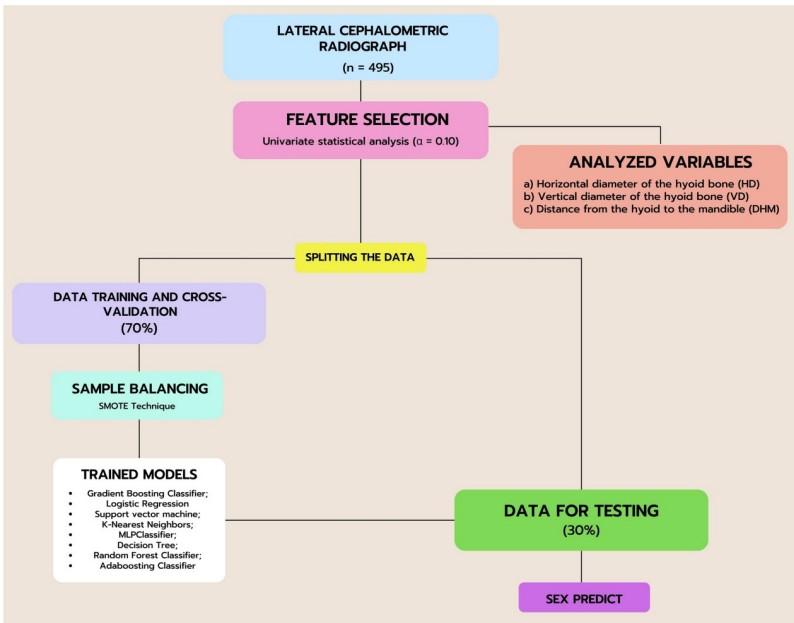

**Fig 2. Flowchart diagram illustrating the data analysis process using machine learning models.**

training data [22]. This technique aims to increase the representation of the minority class, thereby improving the performance of the predictive model. The 'imbalanced-learn' library was used to implement this technique. Additionally, the independent variables were normalized to standardize the data before training and testing the models, ensuring a comparable scale across different variables.

To assess the generalization capability of the models to unseen data, cross-validation technique was employed by dividing the data into k subsets and training the model k times. In each iteration, k-1 subsets were used for training, and the remaining subset was reserved for validation. The average performance of the cross-validation was calculated using a 5-fold scheme. A flowchart detailing all the steps for model construction can be seen in Fig 2.

## Metrics and model evaluation

The evaluation of the distinction between the different classes for each model was performed using the Receiver Operating Characteristic (ROC) curve and calculating the Area Under the Curve (AUC). This involved analyzing the rates of false positives and true positives at various classification thresholds using the 'roc_curve' function from the 'scikit-learn' package. The test data were used with actual labels and predicted probabilities for the positive class from the model. To calculate the AUC, the 'roc_auc_score' function from the same package was employed, providing a measure of the model's discriminative power. Plots were generated using the 'matplotlib.pyplot' library.

Furthermore, metrics such as accuracy, recall, precision, and F1 Score were calculated for each model. For all metrics utilized, 95% confidence intervals (CI95%) were computed. Bootstrap technique with 1,000 iterations employing random sampling with replacement was applied to the test data, setting CI95% as the 2.5th and 97.5th percentiles of the bootstrap distribution of metrics. For 5-fold cross-validation CI95% calculation, the model underwent evaluation on diverse training and test subsets, with metrics computed for each fold. This

facilitated direct computation of CI95% for cross-validation performance metrics, showcasing model performance variability across distinct data partitions. The bootstrap technique was also used for performance comparison among different algorithms. The mean difference between the obtained AUCs was calculated, and the 95% confidence interval (CI95%) of this difference was determined. A statistically significant difference was considered when zero was not included in the calculated confidence intervals.

The assessment of variable importance for each predictive model was conducted using the 'feature_importances_' function from the Scikit-learn library. This function allows for a graphical evaluation of the relevance of each variable for prediction, indicating its impact on the model. It is important to note that the KNN, SVM, and MLP models do not directly support this function due to their specific characteristics.

## Results

A total of 495 patients were included in the sample, divided into 41.4% men and 58.6% women, with a mean age of 36.6 ± 12.4 years (36.4 ± 11.8 for men and 36.7 ± 12.9 for women). Univariate analyses indicated that all hyoid bone variables showed statistical significance ($p < 0.05$), and therefore, they were included in the predictive model. The effect size and test power for each comparison can be seen in Table 1.

The horizontal length of the hyoid bone demonstrated greater importance in all models (Fig 3). The performances of the trained predictive models, along with the respective hyperparameters used, are detailed in Table 2.

The predictive models showed AUC values ranging from 0.72 [CI95% = 0.62–0.81] to 0.86 [CI95% = 0.79–0.92] on the test data and from 0.78 [CI95% = 0.73–0.84] to 0.84 [CI95% = 0.75–0.92] in cross-validation, indicating high predictive power (Fig 4). The algorithms that stood out were logistic regression and random forest classifier, which achieved the highest AUC values, with the latter also obtaining the highest accuracy rates (Table 2). When comparing the different predictive models, the Decision Tree algorithm showed the poorest performance among the evaluated algorithms, with no statistically significant difference only when compared to the KNN model (Fig 5).

## Discussion

Sex identification is important in numerous fields such as anthropology and forensic science, particularly crucial when dealing with human skeletal remains [3, 23]. Sexual dimorphism can

**Table 1. Measurements of hyoid bone variables by sex.**

| Measurement | | Mean ± SD | p-value* | Effect size** | Power Test |
|---|---|---|---|---|---|
| HL | | | | | |
| | Male | 37.6 ± 4.09 | <0.001 | 1.177 | 0.99 |
| | Female | 33.1 ± 3.62 | | | |
| VL | | | | | |
| | Male | 5.43 ± 3.06 | 0.026 | 0.204 | 0.69 |
| | Female | 4.94 ± 1.74 | | | |
| DHM | | | | | |
| | Male | 21.4 ± 6.26 | <0.001 | 0.556 | 0.99 |
| | Female | 18.1 ± 5.56 | | | |

\* p-value of the Student's T test for independent samples;

\** Effect size calculated by Cohen's d;

Feature Importance

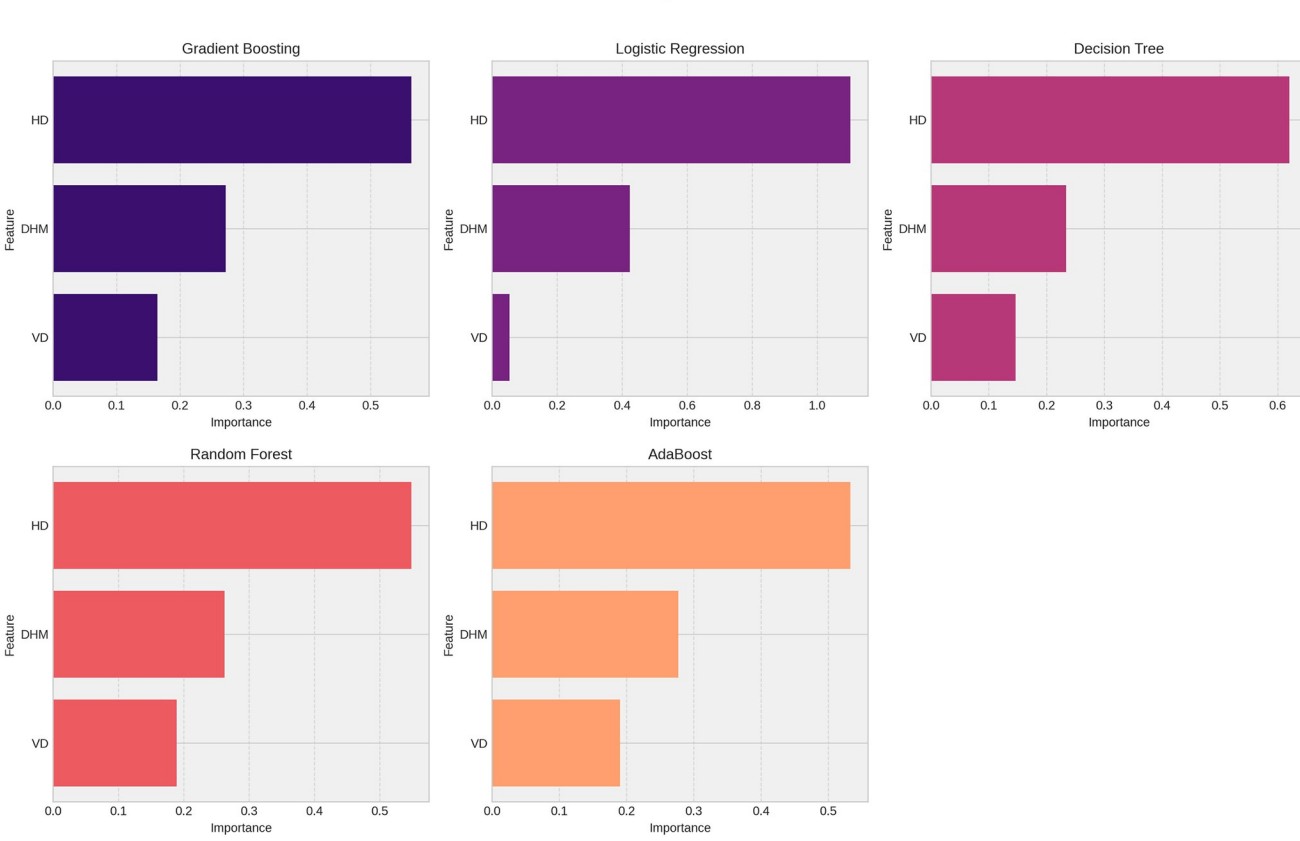

**Fig 3. Results of feature importance analysis from machine learning models.**

be observed in craniofacial complex structures [10, 24–27], and in this regard, the hyoid bone has shown good predictive capability for sex determination [13, 15, 16, 28, 29]. Therefore, the present study aimed to develop a supervised machine learning model for sex determination based on hyoid bone measurements, using features extracted from two-dimensional lateral cephalometric radiograph images. This method proved to have excellent predictive capacity, demonstrating utility in estimating sex based solely on hyoid bone measurements.

While other craniofacial measures such as the frontal bone, dental measurements, mandibular and cervical vertebral measurements have been used to assess sexual dimorphism, studies using hyoid bone measurements for sex prediction through supervised machine learning algorithms are scarce in the literature [10, 24–27]. Despite traditional techniques involving conventional statistical methods, this study adopted an innovative approach by combining hyoid bone measurements with supervised machine learning to explore this topic. Research using different methodologies has concluded that the hyoid bone is sexually dimorphic, capable of aiding physical anthropologists in sex determination [13, 15, 16]. As a result, male hyoid bone structures are generally larger than female hyoid bones in nearly all dimensions, especially in length and total width [28]. Mukhopadhyay et al. (2010) similarly observed that the hyoid bone in the male population was wider compared to females [15]. This aligns with the findings of this study, which demonstrated that hyoid bone dimensions showed a larger average size across all analyzed variables.

**Table 2. Summary of the metrics obtained during the cross-validation and test phases of the models, along with their optimal hyperparameters.**

| Model | Optimal Hyperparameters | Cross-validation results [CI95%] | Test Data Results [CI95%] |
|---|---|---|---|
| Logistic Regression | C: 0.1 | Accuracy = 0.77 [0.73–0.83] | Accuracy = 0.78 [0.70–0.85] |
| | max_iter: 50 | Precision = 0.77 [0.73–0.83] | Precision = 0.79 [0.73–0.86] |
| | penalty: l1 | Recall = 0.77 [0.70–0.85] | Recall = 0.78 [0.70–0.85] |
| | l1_ratio: 0.2 | F1-Score = 0.77 [0.71–0.85] | F1-Score = 0.78 [0.71–0.85] |
| | solver: liblinear | | |
| Gradient Boosting Classifier | n_estimators: 200 | Accuracy = 0.78 [0.75–0.85] | Accuracy = 0.75 [0.68–0.83] |
| | learning_rate: 0.01 | Precision = 0.78 [0.75–0.86] | Precision = 0.76 [0.70–0.83] |
| | max_depth: 5 | Recall = 0.78 [0.75–0.86] | Recall = 0.75 [0.68–0.83] |
| | min_samples_leaf: 4 | F1-Score = 0.78 [0.69–0.83] | F1-Score = 0.75 [0.69–0.83] |
| | min_sample_split: 2 | | |
| K-Nearest Neighbors | n_neighbors: 3 | Accuracy = 0.77 [0.73–0.84] | Accuracy = 0.72 [0.65–0.80] |
| | weights: distance | Precision = 0.77 [0.73–0.84] | Precision = 0.75 [0.68–0.82] |
| | leaf_size: 1 | Recall = 0.77 [0.73–0.84] | Recall = 0.72 [0.65–0.80] |
| | p: 5 | F1-Score = 0.77 [0.73–0.84] | F1-Score = 0.73 [0.66–0.80] |
| Support Vector Machine | kernel: rbf | Accuracy = 0.77 [0.70–0.82] | Accuracy = 0.74 [0.67–0.81] |
| | C: 23 | Precision = 0.77 [0.70–0.83] | Precision = 0.77 [0.70–0.84] |
| | gamma: auto | Recall = 0.77 [0.70–0.82] | Recall = 0.74 [0.67–0.81] |
| | | F1-Score = 0.77 [0.70–0.82] | F1-Score = 0.75 [0.68–0.81] |
| MLP Classifier | activation: relu | Accuracy = 0.76 [0.70–0.86] | Accuracy = 0.75 [0.68–0.82] |
| | alpha: 0.01 | Precision = 0.76 [0.70–0.86] | Precision = 0.76 [0.70–0.83] |
| | hidden_layer_sizes: 1000 | Recall = 0.76 [0.70–0.86] | Recall = 0.75 [0.68–0.82] |
| | learning_rate_init: 0.1 | F1-Score = 0.76 [0.70–0.86] | F1-Score = 0.75 [0.68–0.82] |
| | max_iter: 1000 | | |
| | solver: sgd | | |
| Decision Tree | criterion: gini | Accuracy = 0.75 [0.70–0.80] | Accuracy = 0.77 [0.70–0.84] |
| | max_depth: 5 | Precision = 0.75 [0.70–0.80] | Precision = 0.81 [0.74–0.86] |
| | splitter: best | Recall = 0.75 [0.70–0.80] | Recall = 0.77 [0.70–0.84] |
| | | F1-Score = 0.75 [0.70–0.80] | F1-Score = 0.77 [0.70–0.84] |
| Random Forest Classifier | max_depth: 20 | Accuracy = 0.79 [0.73–0.85] | Accuracy = 0.76 [0.69–0.83] |
| | n_estimators: 200 | Precision = 0.79 [0.73–0.85] | Precision = 0.77 [0.70–0.84] |
| | min_samples_split: 10 | Recall = 0.79 [0.73–0.85] | Recall = 0.77 [0.69–0.83] |
| | min_samples_leaf: 4 | F1-Score = 0.79 [0.73–0.85] | F1-Score = 0.77 [0.70–0.83] |
| | criterion: gini | | |
| | max_features: auto | | |
| Adaboost Classifier | base_estimator_max_depth: 3 | Accuracy = 0.79 [0.71–0.84] | Accuracy = 0.76 [0.74–0.87] |
| | n_estimators: 200 | Precision = 0.79 [0.72–0.84] | Precision = 0.77 [0.75–0.87] |
| | base_estimator_min_samples_split: 2 | Recall = 0.79 [0.71–0.84] | Recall = 0.77 [0.74–0.87] |
| | base_estimator_min_samples_leaf: 1 | F1-Score = 0.79 [0.71–0.84] | F1-Score = 0.77 [0.74–0.87] |
| | learning_rate: 0.01 | | |

Horizontal hyoid bone size demonstrated the highest predictive capacity, and was the most important variable in the tested models. While hyoid bone shape did not exhibit strong predictive capacity for determining sex, some indications were found, though without significant difference for sexual dimorphism [30]. However, osteometric measurements showed pronounced sexual dimorphism in the hyoid bone [31], with male hyoid bones being larger than those of females. Additionally, there is a significant difference in the length of the hyoid

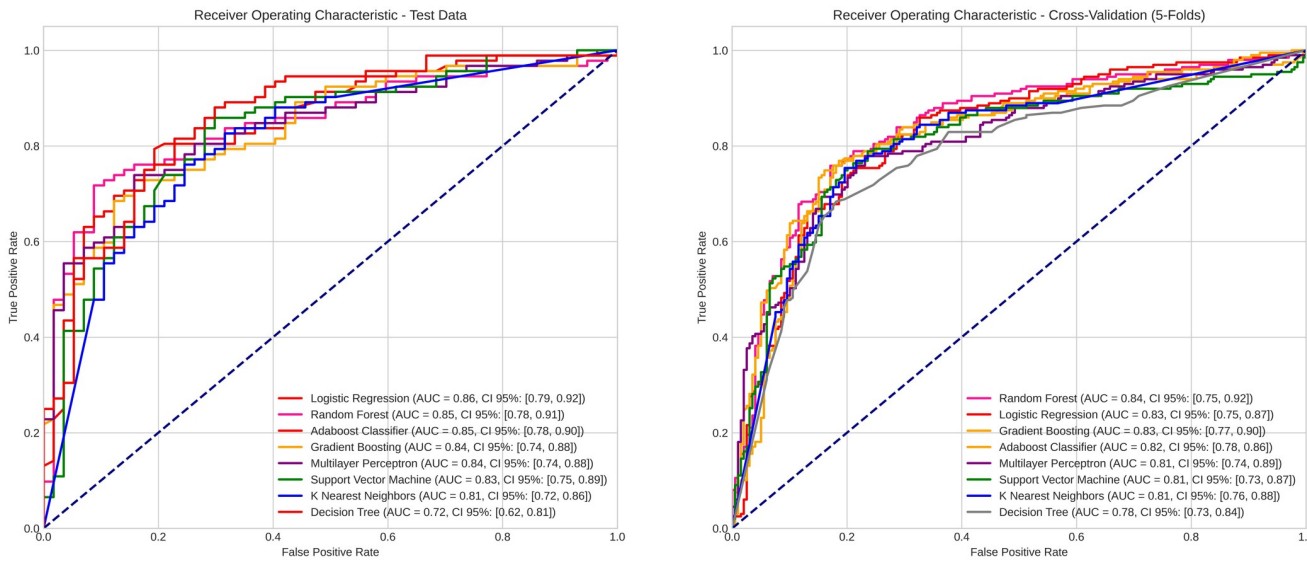

**Fig 4. Evaluation of classification models using ROC curves, for testing and cross-validation.**

body at the midline [13]. To minimize limitations, only linear measurements were used in this study to ensure method reproducibility. Furthermore, only adults with complete craniofacial development were included, eliminating variations in hyoid bone size across different age groups [27]. The use of linear measurement methods and exclusion of children and adolescents from the sample ensured greater accuracy in results related to hyoid bone sexual dimorphism.

Sex determination and identification of population origin are essential elements in forensic investigation. Unlike the pelvic bone, the main challenge with the skull is that sexual dimorphism of craniofacial complex structures varies among different population groups [32]. Previous studies using discriminant analysis found AUCs greater than 0.8 in populations from India and Turkey using hyoid bone measurements [20, 21]. Despite consensus on hyoid bone sexual dimorphism and larger dimensions associated with males, there is a lack of studies

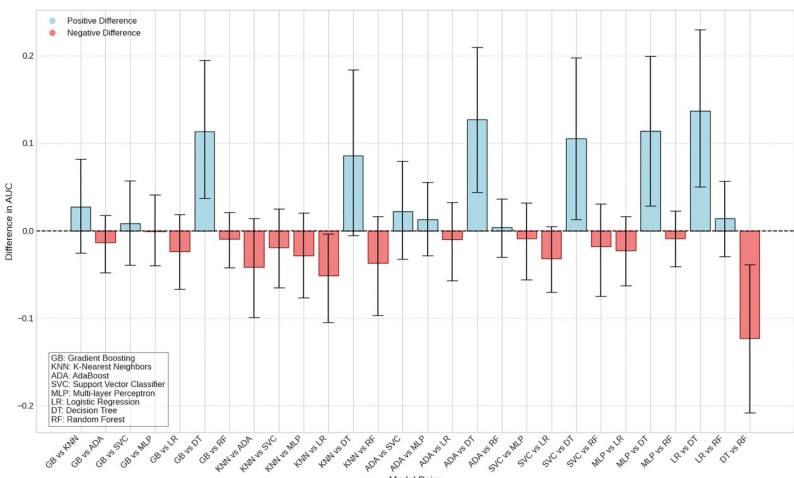

**Fig 5. Difference in AUC between model pairs with 95%CI.**

using artificial intelligence-based methods for sex determination. To the best of our knowledge, this study is the first to apply such methods to a sample composed of Brazilians. It is important to highlight, however, that this study was conducted exclusively on a specific Brazilian population, making its application limited to this group. Replication in different ethnic groups is necessary to validate these findings.

Several previous studies have extensively examined hyoid bone size for sex estimation, but relied on skeletons for measurement [13, 15, 16]. Conventional two-dimensional lateral cephalometric analysis presents challenges in accurately identifying measurement points due to overlapping bone structures, which may be considered a limitation of this study. However, the method showed good reproducibility. The decision to use a 14-day interval for repeating measurements during evaluator calibration was based on previous studies that employed the same interval [5, 33–35]. Although there is a potential risk of memory retention bias due to the short interval, this risk is minimized in this study because the analyzed variables were numerical, making memorization less likely. Additionally, other studies involving machine learning-based methods have also used variables obtained from two-dimensional lateral cephalometric analyses to estimate sex [5, 36]. Thus, the method can be used as an auxiliary tool in sex determination in forensic and anthropological fields. The application of machine learning techniques offers an innovative approach with good accuracy, potentially overcoming some limitations of conventional statistical analyses. This makes the technique promising for future studies aiming to enhance accuracy in sex identification across different population contexts.

## Conclusions

The hyoid bone demonstrated sexual dimorphism, especially in the horizontal dimension. The supervised machine learning model showed good predictive accuracy, indicating hyoid bone measurements potential for sex determination in forensic and anthropological contexts. However, this method should be considered as an adjuvant tool and not as a sole alternative for sex estimation, except when no other resources are available. These findings suggest that the application of artificial intelligence methods can enhance the accuracy of sex estimation, contributing to significant advancements in the field. Future studies should replicate this method in different ethnic groups to validate and generalize the results obtained.

## Author Contributions

**Conceptualization:** Aline Xavier Ferraz, Ângela Graciela Deliga Schroder, Rosane Sampaio Santos, Cristiano Miranda de Araujo.

**Data curation:** Aline Xavier Ferraz, Flavio Magno Gonçalves, Ana Paula Testa Pezzin.

**Formal analysis:** Aline Xavier Ferraz, Flavio Magno Gonçalves, Ana Paula Testa Pezzin, Karinna Verissimo Taveira, Cristiano Miranda de Araujo.

**Funding acquisition:** Ana Paula Testa Pezzin, Allan Abuabara, Flares Baratto-Filho.

**Investigation:** Aline Xavier Ferraz.

**Methodology:** Bianca Simone Zeigelboim, Karinna Verissimo Taveira, Flares Baratto-Filho.

**Project administration:** Rosane Sampaio Santos.

**Resources:** Allan Abuabara, Cristiano Miranda de Araujo.

**Supervision:** Erika Calvano Küchler, Bianca Simone Zeigelboim, Cristiano Miranda de Araujo.

**Validation:** Erika Calvano Küchler.

**Visualization:** Ângela Graciela Deliga Schroder, Karinna Verissimo Taveira.

**Writing – original draft:** Aline Xavier Ferraz, Allan Abuabara, Cristiano Miranda de Araujo.

**Writing – review & editing:** Ângela Graciela Deliga Schroder, Flavio Magno Gonçalves, Erika Calvano Küchler, Rosane Sampaio Santos, Bianca Simone Zeigelboim, Flares Baratto-Filho.

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
