## [Decision Letter · Decision Letter 0]

27 Aug 2024

PONE-D-24-27275ARTIFICIAL INTELLIGENCE MODEL FOR PREDICTING SEXUAL DIMORPHISM THROUGH THE HYOID BONE IN ADULT PATIENTSPLOS ONE

Dear Dr. Baratto-Filho,

Thank you for submitting your manuscript to PLOS ONE. After careful consideration, we feel that it has merit but does not fully meet PLOS ONE’s publication criteria as it currently stands. Therefore, we invite you to submit a revised version of the manuscript that addresses the points raised during the review process.

We look forward to receiving your revised manuscript.

Kind regards,

Johari Yap Abdullah, B.S. & I.T, GradDip ICT, M.Sc, Ph.D.

Academic Editor

PLOS ONE

Journal Requirements:

Reviewers' comments:

Reviewer's Responses to Questions

**Comments to the Author**

1. Is the manuscript technically sound, and do the data support the conclusions?

Reviewer #1: Yes

Reviewer #2: Yes

2. Has the statistical analysis been performed appropriately and rigorously? 

Reviewer #1: I Don't Know

Reviewer #2: Yes

3. Have the authors made all data underlying the findings in their manuscript fully available?

Reviewer #1: No

Reviewer #2: Yes

4. Is the manuscript presented in an intelligible fashion and written in standard English?

Reviewer #1: Yes

Reviewer #2: Yes

5. Review Comments to the Author

Reviewer #1: COMMENTS FOR THE AUTHOR:

Dear authors,

The paper submitted to the journal addresses the topic of “Artificial intelligence model for predicting sexual dimorphism through the hyoid bone in adult patients”

The outcomes of the study are quite confirmatory, but the good study design deserves attention and a chance to reach the readers.

I have a few comments:

1. In introduction, paragraph 2, please add reference for “This bone can exhibit morphological sexual differences, in which men have a slightly larger and more robust hyoid bone, while women have smaller dimensions”

2. In introduction paragraph 3, do you mean to say there are no previous studies conducted on sexual dimorphism of hyoid bone using machine learning? If there is scarce data, need to add some details on the limited articles published and why do you still feel the need to conduct this study.

3. In introduction add some detail what other craniofacial bones are used for sex prediction

4. There is no information on the sample size calculated and the final sample size for this study in methodology section.

5. Any software used for ICC calculation?

6. It would be better to use the term “length” instead of the word “diameter” for the variables as you are measuring length from one point to another rather than the actual diameter.

7. In results section, you have changed the terminology from horizontal diameter to horizontal length. Please use one term throughout the text.

8. There is inconsistency in the reference style with some displaying full journal names and some with abbreviations only. Check with the journal guidelines to adjust the reference list

9. Do you consider an AUC of 0.78 to 0.84 enough for sexual dimorphism in the forensic field? It is almost one wrong case out of every 5. Explain that this method is adjuvant and should be considered as a sole alternative only when no other resource is available.

10. Can you conclude that these prediction models are for Brazilian ethnic group only? Please highlight this in discussion

11. Please use high resolution images

Reviewer #2: Dear authors,

The article is sound and can be accepted as it is with minimal modifications. Inclusion of the different methods of analysis in Figure 4 and Figure 5 is appreciated. The interclass correlation (ICC) >0.8 after 14-day interval can be accepted, though preferably 1-3 months would be better than 14 days to avoid memory retention of the specimens. Justification of the short interval could be included in the discussion section.

6. PLOS authors have the option to publish the peer review history of their article (what does this mean?). If published, this will include your full peer review and any attached files.

Reviewer #1: No

Reviewer #2: No

---

## [Author Response · Author response to Decision Letter 0]

28 Aug 2024

Ref: Submission ID PONE-D-24-27275

Dear Johari Yap Abdullah,

Academic Editor,

PLOS ONE

Thank you very much for your message. We are submitting a revised version of our manuscript after addressing the areas of concern mentioned. The changes to the manuscript are highlighted in red within the document. Please find below our responses to the points raised in your email.

We hope that our corrections are appropriate and that the manuscript may now be reconsidered for publication. Should you have further questions or requests, please do not hesitate to contact us.

Yours sincerely,

Dr. Flares Baratto-Filho

Reviewer Comments:

Reviewer #1 - The paper submitted to the journal addresses the topic of “Artificial intelligence model for predicting sexual dimorphism through the hyoid bone in adult patients”. The outcomes of the study are quite confirmatory, but the good study design deserves attention and a chance to reach the readers.

1. In introduction, paragraph 2, please add reference for “This bone can exhibit morphological sexual differences, in which men have a slightly larger and more robust hyoid bone, while women have smaller dimensions”

Answer: We appreciate the comment, and the reference for this section of the text has been added.

2. In introduction paragraph 3, do you mean to say there are no previous studies conducted on sexual dimorphism of hyoid bone using machine learning? If there is scarce data, need to add some details on the limited articles published and why do you still feel the need to conduct this study.

Answer: We appreciate the comment, and the text has been clarified to include the justification for conducting this new study, highlighting the scarcity of available studies and the need to address this gap in the literature.

3. In introduction add some detail what other craniofacial bones are used for sex prediction

Answer: We appreciate the comment, and information about other craniofacial bones used for sex prediction in forensic science has been added to the introduction.

4. There is no information on the sample size calculated and the final sample size for this study in methodology section. 

Answer: We appreciate the comment. The information about the sample size calculation has been added to the methodology section of the text.

5. Any software used for ICC calculation?

Answer: We appreciate the comment, and the software used for the ICC calculation has been added to the text.

6. It would be better to use the term “length” instead of the word “diameter” for the variables as you are measuring length from one point to another rather than the actual diameter.

Answer: We appreciate the comment, and the suggestion has been accepted. The terminology has been standardized to 'length' throughout the text.

7. In results section, you have changed the terminology from horizontal diameter to horizontal length. Please use one term throughout the text.

Answer: We appreciate the comment, and the suggestion has been accepted. The terminology has been standardized to 'length' throughout the text.

8. There is inconsistency in the reference style with some displaying full journal names and some with abbreviations only. Check with the journal guidelines to adjust the reference list.

Answer: We appreciate the comment. The journal names in the reference section have been reviewed and standardized according to the guidelines recommended by the journal.

9. Do you consider an AUC of 0.78 to 0.84 enough for sexual dimorphism in the forensic field? It is almost one wrong case out of every 5. Explain that this method is adjuvant and should be considered as a sole alternative only when no other resource is available.

Answer: We appreciate the comment. As suggested, we clarified in the conclusion of the article that this method should be considered an adjuvant tool and not as a sole alternative for sex estimation, being recommended as the primary resource only in the absence of other options.

10. Can you conclude that these prediction models are for Brazilian ethnic group only? Please highlight this in discussion.

Answer: We appreciate the comment, and this has been highlighted in the discussion.

11. Please use high resolution images

Answer: We appreciate the feedback. All images were submitted at 600 dpi; however, it is possible that the resolution was reduced by the submission platform when generating the proof for the reviewers. We will look into this issue and ensure that the images are of the appropriate resolution in the final version of the manuscript.

Reviewer #2 - The article is sound and can be accepted as it is with minimal modifications. 

1. Inclusion of the different methods of analysis in Figure 4 and Figure 5 is appreciated. The interclass correlation (ICC) >0.8 after 14-day interval can be accepted, though preferably 1-3 months would be better than 14 days to avoid memory retention of the specimens. Justification of the short interval could be included in the discussion section.

Answer: We appreciate the comment, and the suggestion was carefully considered. We have added a justification for the choice of the 14-day interval to the Discussion section, highlighting the factors that influenced this decision.

---

## [Decision Letter · Decision Letter 1]

8 Sep 2024

ARTIFICIAL INTELLIGENCE MODEL FOR PREDICTING SEXUAL DIMORPHISM THROUGH THE HYOID BONE IN ADULT PATIENTS

PONE-D-24-27275R1

Dear Dr. Baratto-Filho,

We’re pleased to inform you that your manuscript has been judged scientifically suitable for publication and will be formally accepted for publication once it meets all outstanding technical requirements.

Kind regards,

Johari Yap Abdullah, B.S. & I.T, GradDip ICT, M.Sc, Ph.D.

Academic Editor

PLOS ONE

Additional Editor Comments (optional):

Reviewers' comments:

Reviewer's Responses to Questions

**Comments to the Author**

1. If the authors have adequately addressed your comments raised in a previous round of review and you feel that this manuscript is now acceptable for publication, you may indicate that here to bypass the “Comments to the Author” section, enter your conflict of interest statement in the “Confidential to Editor” section, and submit your "Accept" recommendation.

Reviewer #1: All comments have been addressed

Reviewer #2: All comments have been addressed

2. Is the manuscript technically sound, and do the data support the conclusions?

Reviewer #1: Yes

Reviewer #2: Yes

3. Has the statistical analysis been performed appropriately and rigorously? 

Reviewer #1: Yes

Reviewer #2: Yes

4. Have the authors made all data underlying the findings in their manuscript fully available?

Reviewer #1: Yes

Reviewer #2: Yes

5. Is the manuscript presented in an intelligible fashion and written in standard English?

Reviewer #1: Yes

Reviewer #2: Yes

6. Review Comments to the Author

Reviewer #1: (No Response)

Reviewer #2: Dear Authors,

Thank you for a mending the article. It is now fit for publications without any modifications.

Regards.

7. PLOS authors have the option to publish the peer review history of their article (what does this mean?). If published, this will include your full peer review and any attached files.

Reviewer #1: No

Reviewer #2: No

---

## [Editor Report · Acceptance letter]

10 Oct 2024

PONE-D-24-27275R1 

PLOS ONE

Dear Dr. Baratto-Filho, 

I'm pleased to inform you that your manuscript has been deemed suitable for publication in PLOS ONE. Congratulations! Your manuscript is now being handed over to our production team.

Kind regards, 

on behalf of

Dr. Johari Yap Abdullah 

Academic Editor

PLOS ONE